# Fungal Infections in Critically Ill COVID-19 Patients: Inevitabile Malum

**DOI:** 10.3390/jcm11072017

**Published:** 2022-04-04

**Authors:** Nikoletta Rovina, Evangelia Koukaki, Vasiliki Romanou, Sevasti Ampelioti, Konstantinos Loverdos, Vasiliki Chantziara, Antonia Koutsoukou, George Dimopoulos

**Affiliations:** 1st Department of Respiratory Medicine, Medical School, National and Kapodistrian University of Athens and “Sotiria” Chest Disease Hospital, 152 Mesogeion Ave, 11527 Athens, Greece; e.koukaki@yahoo.gr (E.K.); vassoromanou@gmail.com (V.R.); sevi.ampelioti@gmail.com (S.A.); kloverdos@yahoo.com (K.L.); vchantziara@yahoo.gr (V.C.); koutsoukou@yahoo.gr (A.K.); gdimop@med.uoa.gr (G.D.)

**Keywords:** fungal infections, critically ill, CAPA, COVID-19, CAM, CAC

## Abstract

Patients with severe COVID-19 belong to a population at high risk of invasive fungal infections (IFIs), with a reported incidence of IFIs in critically ill COVID-19 patients ranging between 5% and 26.7%. Common factors in these patients, such as multiple organ failure, immunomodulating/immunocompromising treatments, the longer time on mechanical ventilation, renal replacement therapy or extracorporeal membrane oxygenation, make them vulnerable candidates for fungal infections. In addition to that, SARS-CoV2 itself is associated with significant dysfunction in the patient’s immune system involving both innate and acquired immunity, with reduction in both CD4^+^ T and CD8^+^ T lymphocyte counts and cytokine storm. The emerging question is whether SARS-CoV-2 inherently predisposes critically ill patients to fungal infections or the immunosuppressive therapy constitutes the igniting factor for invasive mycoses. To approach the dilemma, one must consider the unique pathogenicity of SARS-CoV-2 with the deranged immune response it provokes, review the well-known effects of immunosuppressants and finally refer to current literature to probe possible causal relationships, synergistic effects or independent risk factors. In this review, we aimed to identify the prevalence, risk factors and mortality associated with IFIs in mechanically ventilated patients with COVID-19.

## 1. Introduction

COVID-19, caused by SARS-CoV2, made its appearance at the end of 2019 in Wuhan (China) and rapidly spread worldwide, evolving to an emergency global public health event. This is justified by the fact that, to date, over 250 million people have been infected worldwide, and more than 5 million have died [1]. By the beginning of March 2020, the World Health Organization (WHO) officially labeled the disease as a pandemic [2]. The disease has a wide range of symptoms; patients may present asymptomatic, with mild symptoms such as fever and cough or worse, such as severe cases developing dyspnea and hypoxia [3,4]. Like in severe acute respiratory syndrome coronavirus (SARS-CoV) and Middle East respiratory syndrome (MERS), SARS-CoV2 may induce a hyper-inflammatory response (cytokine storm) and a pneumonia complicated by Acute Respiratory Distress Syndrome (ARDS), severe alveolar damage and inflammatory exudation [5,6].

In COVID-19, primary coinfections are rare [7,8]; however, critically ill patients (which represent around 20% of all patients) and especially those who are finally admitted to the Intensive Care Unit (ICU) (around 5%) are more vulnerable to develop secondary infections on the basis of multiple organ failure, prolonged time on mechanical ventilation and dependence on renal replacement therapy or extracorporeal membrane oxygenation [9,10]. In addition, SARS-CoV2 itself is associated with significant dysfunction in the patient’s immune system involving both innate and acquired immunity, with reduction in both CD4^+^ T and CD8^+^ T lymphocyte counts. A cytokine storm takes place, which is characterized by excessively increased pro-inflammatory molecules, inhibition of natural killer cells and cytotoxic lymphocytes [11,12]. On this ground, secondary infections as a complication of viral respiratory diseases are not uncommon, and they have been described in previous pandemics.

Infections in these patients need to be identified early because they affect the management, and more significantly, the outcome. Fungal infections are especially associated with a higher mortality rate in critically ill patients admitted to the ICU, raising a considerable concern about invasive fungal infections in COVID-19 patients. Interestingly, in an Italian ICU, the incidence of secondary blood stream infections (BSIs) in COVID-19 patients was almost 20 times higher than the incidence reported in European ICUs in non COVID-19 patients [13]. According to authors, this finding could be attributed to factors such as the dysregulated immune system in severe COVID-19, the extensive use of antimicrobials in these patients, as well as, the worse adherence to the infection control and prevention measures due to the overwhelming pandemic wave in Italy at that time.

In this review, we provide an up to date insight on the epidemiology and the risk factors for invasive fungal infections, as well as, the current view on the potential role of drugs which modulate and/or compromise immune response in increasing the prevalence of fungal co-infections in critically ill COVID-19 patients.

## 2. Pathophysiology and Risk Factors

The emergence of fungal co-infections in COVID-19 patients is not that unexpected considering the MERS and SARS outbreaks [14]. The variety of fungal co-infections (most commonly pulmonary or tracheobronchial aspergillosis—CAPA, CAC and CAM) and their relatively high incidence, as described above, especially in critically ill patients, have posed questions on what the underlying pathogenetic mechanisms are of such an occurrence and whether any risk factors could be identified [15].

The SARS-CoV2 spike protein binds to angiotensin converting enzyme 2 (ACE2) receptor of epithelial cells and type 2 pneumocytes, thus allowing viral entry. The release of danger-associated molecular patterns (DAMPs) by dying or damaged cells ignites an immune response and a cascade of inflammation, which in turn leads to tissue damage [16,17,18]. This extensive lung damage may lead to higher vulnerability to invasive fungal infections such as pulmonary aspergillosis [14]. DAMPs have been implicated in the regulation of inflammation of fungal diseases, and host inflammation may favor the transition of fungal colonization to fungal infection [19]. This could explain in part why patients with hyperinflammatory response (such as COVID-19 critically ill patients) are more vulnerable to fungal infections, Figure 1.

On the other side of the immunologic spectrum, COVID-19 interferes toward impaired local immune response, dysfunctional mucociliary activity and disruption of epithelial barriers [17,20]. In the case of *Aspergillus*, conidia in the airways are cleared poorly, enabling bronchial inflammation and invasion and possibly leading to CAPA [20]. The high prevalence of CAPA but with low blood galactomannan detection may indicate less frequent angioinvasion compared to usual invasive asperillosis and maybe local disease [21]. Lymphopenia—also a common lab characteristic of COVID-19 infection—and, consequently, possibly T-cell lymphocyte population decline may lead to a favorable environment for invasive fungal infections [16]. The effect of immunomodulation is under debate, and it will be discussed separately.

In general, it could be hypothesized that the high incidence of aspergillosis and candidemia in COVID-19 patients could be related to high rates of invasive procedures, such as intubation, which may predispose to fungal colonization and proliferation; prolonged/high corticosteroid or other immunomodulatory treatment regimens; underlying (or developing due to COVID-19) pulmonary disease; dysregulation of the immunity caused by COVID-19; and empiric antimicrobial therapy changing the flora/microbiota of the respiratory tract [14,17].

The risk factors can be divided into the following categories.

### 2.1. Host Factors

Initially, for patients with severe COVID-19, it could be hypothesized that increased fungal co-infections correspond to an increased prevalence of pre-existing risk factors for fungal infections (as they are defined by European Organization for Research and Treatment of Cancer and the Mycoses Study Group Education and Research Consortium—EORTC/MSGERC) [15]. However, it seems that only a minority of patients have such risk factors [14,21]. Even though, in general, critically ill patients with COVID-19 are usually older or have more comorbidities, a strong correlation with specific characteristics that might predispose to vulnerability to fungal infections (e.g., disease, gender) has not been documented [14,17]. Structural lung diseases might be of importance for the development of CAPA [14,16], as some studies identify a higher incidence in patients with COPD or asthma [22]. In addition, in India, diabetes and diabetic ketoacidosis have been correlated with increased likelihood of mucormycosis in COVID-19 patients, but not in patients with CAPA [14,21]. Finally, from a small series in Hungary, patients with CAPA seemed to be preferably older males and in need of oxygen support, suggesting that critical illness might be a risk factor [23].

### 2.2. Healthcare Associated Factors

As with other medical reasons, admission to hospital or ICU increases the incidence of infections. Mechanical ventilation/intubation, ICU length of stay and presence of indwelling catheters have been implicated in the emergence of secondary infections in ICU patients. In addition, empiric antibiotic treatment may shift the microbiota towards increased prevalence of fungal species. In literature, fungal infections seem to be correlated with hospital admission and intubation [17], even though CAPA has been described in patients with various respiratory support [22], such as invasive or non-invasive ventilation. There is also a correlation of length of ICU stay with the development of CAPA [22]. Although renal failure seems not to be a risk factor, renal replacement therapy seems to be associated with the development of fungal infections, such as CAPA, in critically ill COVID-19 patients [22]. To our knowledge, there are no case reports for outpatients with COVID-19 CAPA or CAC.

There is a debate in the literature on whether corticosteroids favor fungal infections or not [21], and this will be discussed further, separately. It is a fact that critically ill patients may have more risk factors independent of COVID-19 that may confound the correlations. Table 1.

## 3. Prevalence of Invasive Fungal Infections in Critically Ill COVID-19 Patients

The reported incidence of invasive fungal infections in critically ill COVID-19 patients ranges between 5% and 26.7% [24,25,26,27,28,29,30,31]. Fekkar et al. found an overall incidence of fungal respiratory complications at 4.8% [24]. Chong et al., in their review of 49 studies on secondary pulmonary infections in COVID-19 cases, noted a 6.4% incidence of fungal infections [25]. In a descriptive study of 99 COVID-19 patients, Chen et al. reported fungal co-infections in 5% of the patients [26]. Moreover, among 52 critically ill patients, Yang et al. noted 3 (5.8%) with fungal infections [27], while Musuuza et al. in their meta-analysis reported a pooled prevalence of fungal co-infections in 4% and fungal superinfections in 8% [28]. Furthermore, in a retrospective study of 140 ICU patients, Bardi et al. reported fungal infections in 15% of them [29]. In this line, White et al. reported a far higher prevalence of 26.7% of invasive fungal infections in critically ill COVID-19 patients [30]. On the other hand, in a meta-analysis of 426 COVID-19 patients who were admitted to the ICU, the overall pooled proportion of fungal co-infection was 0.12 [31] (Table 2).

### 3.1. Candidiasis

The risk of invasive candidiasis is high in patients receiving antibacterials, hemodialysis, parenteral nutrition, undergoing mechanical ventilation and having central venous catheters. All these factors are common in the COVID-19 critical-care patient [32,33]. COVID-19-associated candidiasis (CAC) mostly presents as candidaemia, with *Candida albicans* and *Candida glabrata* being the most frequent pathogens. Candidemia in COVID-19 patients has been increasingly described in the literature. In a US hospital, among patients with COVID-19 admitted to the ICU, 8.9% developed candidemia [34]. A single-hospital study in Brazil reported candidemia incidence during the pandemic to be nearly five times higher than before the pandemic [35]. Seagle et al. studied 251 patients with candidemia, of which 25.5% were positive for SARS-CoV2 [36]. Outbreaks of the multidrug-resistant *Candida auris* have also been reported [37] (Table 2).

### 3.2. Aspergillosis

Back from 2018, Schauwvlieghe et al. developed the modified AspICU criteria to help diagnose influenza associated pulmonary aspergillosis (IAPA), which (in the absence of histology) essentially relies on mycological evidence of *Aspergillus* spp. in the form of a positive bronchoalveolar lavage (BAL) culture or positive galactomannan in serum/BAL [38]. In a retrospective observational study of critically ill COVID-19 patients, although the rate of fungal antigenemia was high (around 50%), *Aspergillus* was not identified in any specimens by culturing. This was associated with the prophylactic antifungal therapy the patients were receiving [39]. Despite differences in the pathogenicity of COVID-19 and influenza, obviously a similar manifestation appeared in the critically ill COVID-19 patients. Not long ago, a panel of experts proposed criteria for a new clinical entity called CAPA (COVID-19-Associated Pulmonary Aspergillosis), a superinfection with high incidence and high mortality. In Koehler et al.’s study of 94 cases, the overall incidence was 22.6% [40]. Studies in France [16,41] identified a similar incidence of CAPA (30%). However, other studies showed lower incidence and variation in prevalence, indicating that further investigation is needed [42] (Table 2).

### 3.3. Mucormycosis

Another significant complication of COVID-19 infection is COVID-19 associated mucormycosis (CAM), mostly in patients with uncontrolled diabetes mellitus or in geographical regions with higher incidences of mucormycosis (e.g., India) [43]. CAM includes patients with acute invasive fungal rhino-orbital mucormycosis, or cavitary pulmonary mucormycosis [44,45]. Meawed et al. isolated Mucor in 8.2% of 197 critically ill COVID-19 patients under mechanical ventilation who developed VAP [46]. Selarka et al., in India, characterized mucormycosis in patients with COVID-19 as an epidemic within a pandemic, as from the 2567 COVID-19 patients admitted to 3 tertiary centers, 47 (1.8%) were diagnosed with mucormycosis [47] (Table 2).

### 3.4. Pneumocystis

*Pneumonocystis jirovencii* infections have been reported in low numbers and mainly in patients with other underlying conditions (e.g., HIV, hematologic malignancy) [48]. COVID-19 and *Pneumocystis* share numerous overlapping characteristics, such as radiological, clinical and laboratory findings. As a result, there is a great possibility of misdiagnosis [49]. Alanio et al. studied samples from 108 HIV-negative COVID-19 patients. *P. jirovencii* was postitive in 9.3% [50] of the patients. Blaize et al. also explored the incidence of *Pneumocystis* in pulmonary specimens obtained from severe COVID-19 patients and found it around 1.4% [51]. In March 2021, the first case of *Pneumocystis* confirmed through autopsy was published in a 52-year-old male, who was diagnosed with COVID-19 posthumously [52].

Other rare invasive fungal diseases that have been diagnosed in COVID-19 patients, include *Rhodotorula fungaemia*, *Fusarium* and *Trichosporon* infections [53,54]. *Cryptococcus* should be considered in high-risk patients [55]. Endemic fungal species such as *Histoplasma*, *Coccidioides* or *Blastomyces* should be considered in specific geographical areas (Table 2).


jcm-11-02017-t002_Table 2Table 2Prevalence of Invasive Fungal Infections in critically ill COVID-19 patients.LiteratureTrial Design/PopulationType of IFIIncidenceFekkar A. et al. [24]R, SC, *n* = 145 COVID-19 ICU MV pts screened for fungal superinfection; 54% on ECMOprob/putat IFI(1 *Fusarium* case)4.8%Chong W.H. et al. [25]Literature review; 28 O studies, 21 cr/sSecondary FI6.4%Chen N. et al. [26]R, SC, 99 hospital ptsSecondary FI5%Yang X. et al. [27]R, SC, O, 52 ICU pts
5.8%Musuuza J.S. et al. [28]MA of 118 studiesFungal co- andsuperinfections4% and 8%, respectBardi T. et al. [29]R, SC, 140 ICU ptsFI15%White et al. [30]MC, P, 137 ICU pts screened for IFIIFI26.7%Peng J. et al. [31]SRMA of 9 studiesIFI0.12 (opp)Bishburg E. et al. [34]SC, R, 89 COVID-19 ICU ptsCAC8.9%Nucci M. et al. [35]SCCAC×5 comp toprepandemicSeagle E.E. et al. [36]Surveillance datacandidemiaAmong 251candidemia pts, 25.5% were SARS-CoV-2Gouzien L. et al. [42]R, O, COVID-19 ICU ptsCAPA1.5%Hoenigl M. et al. [43]Review of 80 CAM casesCAM0.3–0.8% prevalence in COVID-19 ICU ptsMeawed T.E. et al. [46]Cross-sectional study of 197 critically-ill MV COVID-19 ptsFungal VAP16.4% *Aspergillus*8.2% mucorSelarka L. et al. [47]P, O, MCCAM1.8%Alanio et al. [50]O, 108 critically-ill COVID-19 ptsPJP9.3%Blaize et al. [51]PCR assays on severe COVID-19 ptsPJP1.4%CAC: COVID-19-associated candidemia, CAM: COVID-19-associated mucormycosis, CAPA: COVID-19-associated pulmonary aspergillosis, Comp to: compared to, Cr/s: case reports/series, (I)FI: (Invasive) fungal infection, MA: metanalysis, MC: multicenter, MV: mechanically ventilated, O: observational, opp: overall pooled proportion, P: prospective, prob: probable, pts: patients, put: putative, R: retrospective, respect: respectively, SC: single center, SRMA: Systematic review and metanalysis.


## 4. Immunosuppressive Therapy as Risk Factor for Fungal Infections in Critically Ill COVID-19 Patients

The emerging question is whether SARS-CoV-2 inherently predisposes to fungal infections in the critically ill patients or the immunosuppressive therapy constitutes the igniting factor for invasive mycoses. To approach the dilemma, one must consider the unique pathogenicity of SARS-CoV-2 with the deranged immune response it provokes, review the well-known effects of immunosuppressants and finally refer to current literature to probe possible causal relationships, synergistic effects or independent risk factors.

As mentioned above, SARS-CoV-2 binds to ACE2 receptors and invades the respiratory epithelium causing mucociliary clearance dysfunction and extensive disruption in mucosal integrity. Moreover, in severely ill patients, the immune response to SARS-CoV-2 is totally dysregulated with defective monocytes and neutrophils, diminished IFN I and III production [56] and a DAMP-driven [57] explosive release of pro-inflammatory cytokines (IL-6, IL-1, IL-2, TNF, MCPI) that further damage the lung [58,59], Figure 1. In addition, the adaptive immunity dysfunction is expressed with lymphocytopenia, which correlates with COVID-19 severity, CD8^+^ and CD4^+^ exhaustion and alterations in Th1 and Th2 responses [60,61].

Current treatment guidelines for critically ill COVID-19 patients target the immune aberration and cytokine storm with corticosteroids and other immunomodulators, such as tocilizumab, Janus kinase (baricitinib) and anakinra [62,63,64]. Glucocorticosteroids (GCS) interfere with virtually total human immunome through transcriptional changes after binding to their intracellular receptors [65]. High doses and long-term use of corticosteroids are known to predispose to bacterial and fungal infections [66]. Tocilizumab, a humanized monoclonic antibody against soluble and membrane IL-6 receptors, used since the mid-1990s in rheumatoid arthritis patients is implicated in a small but significant increase in the risk of infection [67], especially of the respiratory system via the inhibition of Th17 proliferation [68].

All in all, anatomical disruption and immune impairment by SARS-CoV-2 itself together with immunomodulatory therapeutic modalities compose the perfect environment allowing fungi to become invasive [31,69]. Molds and yeasts rarely become pathogenic in immunocompetent hosts. They mostly behave as opportunistic agents [70]. GCS increase susceptibility to invasive aspergillosis by inhibiting macrophage from phagocytosing molds, which in turn germinate quickly into hyphae [71]. IFN type I defects, PMN dysfunction and lymphopenia leave *Aspergillus* uncontrolled to invade damaged respiratory epithelium and cause tracheobronchitis [60,72]. Additional in vitro experiments show that *A. fumigatus* and *A. flavus* thrive in GCS-rich environments [73]. Yeast infections, such as candidaemia—a mostly healthcare-associated infection—are also favored by GCS effects on TNFa, monocytes, macrophages, PMNs and T-lymphocytes [74,75,76,77] and possibly by the decreased expression of human leukocyte antigen DR on the membrane of circulating monocytes [61]. Similarly, GCS are implicated in zygomycosis possibly through phagocyte dysfunction and hyperglycaemia [78,79]. GCS, albeit indispensable in treating *Pneumocystis Jirovecii* pneumonia (PJP) with hypoxemia, constitute a well-recognized risk factor for PJP manifestation, especially during GCS tapering [80,81].

### 4.1. Glucocorticosteroids

Moreover, in the critical care setting, GCS treatment consists of a common predisposing factor for invasive pulmonary Aspergillosis (IPA), as is shown by several observational studies. In a US cohort, 77% of IPA ICU patients received steroids [82]. In the Spanish ICU cohort by Garnacho-Montero et al. [83], steroid treatment (prednisone 20 mg equivalent dose for 3 weeks) was a major predisposing factor for *Aspergillus* colonization and infection. In the ICU special environment, several other factors apart from GCS use predispose to yeast infections: total parenteral nutrition, antibiotic use, intravascular catheters, acute kidney injury, renal replacement therapy, heart disease, mechanical ventilation, prior septic shock, other immunosuppressive medication and underlying comorbidities, such as COPD and diabetes [84,85]. Interestingly, gene polymorphism in ICU population may further increase fungal susceptibility [81,86].

In the COVID-19 era, multiple studies, mostly small, retrospective and observational, highlight an association between GCS and COVID associated Pulmonary Aspergillosis (CAPA) [87,88,89,90,91]. White et al. [30] in one of the largest trials from the UK with prospective screening of 135 PCR-confirmed COVID-19 patients for invasive fungal disease concluded that GCS increased the likelihood of aspergillosis in COVID-19 patients. They implied that negative events in GCS-treated patients could be potentially attributable to CAPA rather than COVID-19 and urged for active surveillance for aspergillosis. Delliere et al. [92], in their French cohort of COVID-19 ICU patients who deteriorated clinically, highlighted a trend with high dose dexamethasone and invasive aspergillosis (11.5 vs. 28.6%, *p* = 0.08, cumulative dose of dexamethasone ≥100 mg OR 3.7, 95% CI 1.0–9.7). The association did not reach significance, probably because of insufficient statistical power. Fekkar et al. [24], in their multicenter trial, emphasized a low incidence of fungal respiratory complications (4.8%) in a population of COVID-19 ICU patients who received GCS less often (16.7%) than in other series. In the multivariate model analysis, GCS were related to invasive pulmonary mold infection (OR 8.55; IQR 6.8–10.3; *p* = 0.01). It is important to note that this correlation referred to GCS received in dose >0.3 mg/kg/day, for >6 weeks, before COVID-19 diagnosis for underlying disease. No patient on GCS, mostly dexamethasone 20 mg/d for 10 days, for COVID-19 developed invasive mycosis. Marr et al. [20] noted that GCS, systemic or inhalational, were the most common immunosuppressive agent among their 20 patients with CAPA. Of note, Meijer et al. [93] compared the incidence of invasive aspergillosis between the first and second wave and correlated the increased incidence of CAPA to the universal introduction of GCS. The significant decrease in the rates of patients needing mechanical ventilation in the second wave (*p* < 0.01) was counterbalanced by a higher percentage of CAPA diagnoses (24.2% vs. 15.2%, second and first wave, respectively, *p* = 0.36). All CAPA patients in their single center series had received dexamethasone. Similarly, Fortarezza et al. [94] published autopsy results from 45 confirmed COVID-19 pts with unusual clinical course; 20% proved to have CAPA (20%). Interestingly, most were patients from the second wave, when GCS were mainstay of treatment.

In the context of Candida infections in COVID-19 patients, Obata et al. [95] in their retrospective chart review of COVID-19 patients compared the incidence of secondary infections between patients who received GCS and those who did not and found a statistically significant increase in fungal infections in patients receiving GCS (12.7% vs. 0.7%, *p* < 0.001). No *Candida* infections vs. 7% incidence were noted among the no-GCS and GCS arms, respectively. In the Brazilian multicenter cohort by Riche et al. [96], comparison between candidemia rates in patients with COVID-19 and non-COVID-19 patients revealed a 10-fold increase in the former group, mainly in those hospitalized in the ICU. Of note, all cases of COVID-related candidemia had received high dose GCS for severe COVID-19. A very well-structured comparison of candidemia in COVID-19 and non-COVID-19 patients [36] found that systemic GCS administration was twice that of non-COVID-19 patients before *Candida* diagnosis. Their data showed that candidemia in patients with COVID-19 appears later than in non-COVID-19 patients, which made them deduce that healthcare practices associated with severe COVID-19, such as immunomodulating treatments, are likely major contributors to candidemia. Steroid use, among others, was mentioned as a risk factor for *Candida auris* bloodstream infection in certain case series [97].

On the contrary, Ho et al. [98] in their multicenter study show similar rates of blood culture positivity for *Candida* in hospitalized COVID-19 patients between GCS and no-GCS groups. Rutsaert et al. observed high incidence of CAPA despite the absence of systemic corticosteroid use [99]. Van Biesen [100] noted a 22% incidence of CAPA in patients not exposed to GCS. Likewise, Wang et al. [101] in their early study showed that treatment with GCS was not an independent risk factor for IPA in COVID-19 ICU patients, as opposed to older age, initial antibiotic usage of β-lactamase inhibitor combination, mechanical ventilation and COPD. Comparably, Janssen et al., in their multinational observational study [102], did not find a correlation between GCS and CAPA.

Concerning infections by *Zygomycetes* in COVID-19 patients, (COVID-19-Associated Mucormycosis, CAM), the cohorts come mainly from the Indian sub-continent. Patel et al. point out that 32.6% of CAM patients had no underlying factor other than SARS-CoV-2 infection, while improper GCS use (higher dose or prolonged administration) was independently associated with the “black fungus disease” [103]. As for tocilizumab, only 2.7% of the CAM patients in the aforementioned study received tocilizumab. Not unexpectedly, the predominant risk factor for mucormycosis was found to be diabetes. The authors suggested that their data should be compared to data from Western countries, where the main predisposing factor for mucormycosis are hematological malignancies and organ transplantation, apart from diabetes mellitus. Several other cohorts reported the frequency of GCS use among CAM patients as high as 60% [104,105], over 85% [106,107] and 100% [107]. With regards to *Pneumocystis jirovecii* Pneumonia (PJP), Chong et al. [108], in their excellent review of PJP-SARS-CoV-2 co-infection case reports, conclude that the virus may predispose to this fungal entity through lymphocytopenia and macrophage dysfunction. Although 10 out of 12 co-infected patients received steroids for COVID-19 management, they mention that a safe conclusion as to the predisposing factors could not be made.

### 4.2. Tocilizumab

As for tocilizumab, there are no data from the pre-COVID era, since there was no indication for ICU patients. In the COVID-era, there are several studies showing the high prevalence of CAPA among tocilizumab-treated patients. Lamoth et al. [109] raised the suspicion of tocilizumab contributing to IPA since the three patients with pulmonary aspergillosis in their cohort of COVID-19 ICU patients had received tocilizumab, as was the case for the non-IPA patients. Kimming et al. [110], in their retrospective study of critically ill COVID-19 patients, showed that receiving tocilizumab was associated with a higher risk of secondary fungal infections (5.6 vs. 0%; *p* = 0.112); of note, more patients in the tocilizumb group received corticosteroids as well. Importantly, the large multinational cohort by Prattes et al. comparing CAPA to non-CAPA COVID-19 critically ill patients [111] demonstrated tocilizumab, and not GCS, as a risk factor for CAPA. Tocilizumab in this study was mostly used in the dose of 8 mg/kg BW.

Reviewing tocilizumab and *Candida* infections, Antinori [112] reported a 6.9% incidence of candidemia among 43 severe COVID-19 patients hospitalized in ward and ICU. All candidemic patients had received TPN and tocilizumab (8 mg/kg repeated within 12 h), while the interval between administration of tocilizumab’s last dose to candidemia was 13 days (median). Seagle et al. examined the receipt of tocilizumab in candidemic patients with and without COVID-19 [36] and found it to be 30 times more common in COVID-19-Associated Candidemia (18.8% vs. 0.5% without COVID-19; *p* ≤ 0.0001). Guaraldi et al. [113] also showed that tocilizumab increases the incidence of candidemia and IFIs. In a Spanish cohort, treatment with tocilizumab alone or in combination with GCS increased the risk of systemic candidiasis (*p*-value = 0.05; 0.010, respectively [114].

These findings are opposed to several smaller case series. Xu et al. [115] traced no secondary fungal infections in a small series of 21 severe and critically ill COVID-19 patients treated with tocilizumab. Obata et al. [95] in the abovementioned trial made a subgroup analysis for tocilizumab treatment together with GCS and revealed no significant difference in fungal infections between patients who received tocilizumab or not. Likewise, Hermine et al. [116], in their prospective, randomized, open-label trial comparing tocilizumab to usual care, found no fungal sepsis cases in the tocilizumab group vs. 2 out of 67 in the usual care group. Recent literature features influenza as independently associated with invasive pulmonary aspergillosis [38]. The same is not, still, the case for COVID-19. The debate is still ongoing whether SARS-CoV-2 predisposes to fungal infections and, if yes, to what extent and how it interacts with the immune modulation by current treatment modalities.

Table 3 summarizes the studies on invasive Aspergillosis in COVID-19 patients.

Table 4 summarizes the studies on Candida infections in COVID-19 patients treated with steroids.

## 5. Mortality and Fungal Infections in Critically Ill COVID-19 Patients

The available data clearly suggest that IFI identification in a critically ill COVID-19 patient confers a highly dismal prognosis, regardless of the causative fungal agent. However, as IFIs tend to occur more frequently to the most seriously affected patients, it is, as of yet, less clear whether this excess mortality associated with IFI diagnosis is attributable to IFI per se, the severity of the underlying disease or, most probably, a combination of both. Unfortunately, a further increase in IFI-driven mortality must be awaited as multi-drug-resistant species, such as *Candida auris*, rapidly spread around the world.

Invasive pulmonary aspergillosis has repeatedly been shown to be associated with increased mortality in critically ill patients with COVID-19 [116]. In an early landmark study, Bartoletti et al. [89] prospectively applied a CAPA screening protocol consisting of consecutive bronchoalveolar lavage (BAL) sampling for galactomannan (GM) measurement in all COVID-19 ICU patients requiring invasive mechanical ventilation (on admission, at day 7 of mechanical ventilation and in case of respiratory deterioration). CAPA was diagnosed in 30 out of 108 enrollees. Mortality was twice as high in CAPA patients compared with their non-CAPA counterparts, and both CAPA diagnosis and higher initial BAL GM levels were identified as independent risk factors for mortality even after adjusting for age, underlying disease severity on ICU admission and need for renal replacement therapy. Although not statistically significant, a trend towards better outcomes was shown in CAPA patients who received treatment with voriconazole. In another prospective study, White and colleagues corroborated the favorable effect of antifungal treatment administration on survival in patients with CAPA [30]. In their CAPA cohort, the clear majority of patients who did not receive appropriate antifungal treatment died, while mortality fell to 46.7% in those properly managed with antifungals. The larger study on CAPA to date was a multinational observational study conducted by the European Confederation of Medical Mycology (ECMM) [111]. 109 ICU patients with histologically proven, probable and possible CAPA according to ECMM/ISHAM diagnostic criteria [30] were analyzed and compared with 483 non-CAPA COVID-19 patients. Length of ICU stay and 90-day mortality were significantly higher in CAPA than in non-CAPA patients (56% versus 41%), and CAPA diagnosis continued to be an independent predictor of mortality after adjusting for age, study center and comorbidities. Collectively, these data strongly support the idea that CAPA may constitute a major contributor to excess mortality in critically ill COVID-19 patients. On the other hand, several reports have described patients diagnosed with CAPA who survived despite lack of appropriate antifungal treatment administration [20,86,117]. Furthermore, in a recently published study on Aspergillus test profiles of 58 patients with CAPA, 30-day ICU mortality was not significantly higher in CAPA patients treated with antifungal agents compared with those left untreated [118]. The authors also failed to show a significant effect of BAL GM positivity or levels on patient outcome, with only serum biomarkers having prognostic value. These findings are directly contradictory to those of earlier studies and may imply that increased fatality rates observed in critically ill patients with CAPA may, at least in part, be attributable to the co-existence of other risk factors for mortality [119].

Candidemia is a usually fatal secondary infection in COVID-19 patients. A case–control study conducted during the first wave of the pandemic showed an extremely high all-cause mortality rate (72.5 versus 26.9%) in COVID-19 ICU patients with candidemia compared with non-candidemic controls matched with CAC cases according to length of hospitalization before candidemia occurrence [120]. Moreover, candidemia appears to be more lethal in critically ill COVID-19 patients in comparison with their non-COVID-19 counterparts. Using data from a nationwide surveillance program in the US, Seagle and colleagues studied 251 patients with candidemia, one quarter of which had been diagnosed with COVID-19 [36]. Hospital all-cause mortality was twice as high in patients with CAC compared with patients without COVID-19 (62.5 vs. 32.1%). In another large study, all patients admitted to the ICUs of a single reference hospital in Turkey during a period of two years extending both before and after the SARS-CoV 2 outbreak were included, and those with *Candida* species isolated in blood cultures were analyzed [121]. Mortality was exceptionally high in both COVID-19 and non-COVID-19 groups, but candidemia-associated fatalities were significantly more common in the former (92.5 vs. 79.4%). A delay in pre-emptive antifungal treatment administration in deteriorating septic patients may have contributed to these overwhelmingly high mortality rates, as one-third of all patients with candidemia included in the study never actually received any antifungal, and this was more common in the COVID-19 group. The same study provides insights into possible risk factors for mortality in CAC. In multivariate logistic regression analysis older age (>65 years), prior corticosteroid administration and presence of sepsis were recognized as potential predictors of adverse outcomes in patients with CAC. In line with the reports, smaller observational studies and case series have consistently documented CAC fatality rates exceeding 50% [122].

An issue of concern with an anticipated effect on mortality is the emergence and rapid spread of resistant *Candida* strains, most notably of the *Candida auris* species, which has preceded the outbreak of SARS-CoV2 but may have been accelerated by the pandemic [123]. Although the only rarely resistant *Candida albicans* remains the most commonly isolated Candida species in candidemic non-COVID-19 and COVID-19 patients alike, occasionally, non-susceptible strains of non-*albicans Candida* species (including *Candida glabrata*, *Candida parapsilosis* and *Candia kruzei)* are increasingly reported and may constitute most bloodstream *Candida* infections in some instances [122,124]. *Candida auris* was originally described in 2009 in a Japanese patient with external otitis and has, thenceforth, been isolated in an ever-growing number of countries across the world [123]. *Candida auris* possesses a unique combination of worrisome features, due to which it has been declared an “urgent threat” by the US Centers for Disease Control and Prevention (CDC) [125]. *Candida auris* is characterized by high rates of resistance to azoles and amphotericin B (85% and 33%, respectively, according to recent CDC data), and pan-resistant strains with additional lack of susceptibility to echinocandins have rarely but increasingly been isolated from patients without prior echinocandin exposure [126]. Furthermore, *Candida auris* can form biofilms, presents difficulties in laboratory identification, is more resilient to commonly used disinfectants and, importantly, can remain on most of the surfaces of the health-care environment in viable forms for long periods of time, the latter facilitating transmission between patients and outbreaks, especially in intensive and long-term care units [123]. Secondary *Candida auris* infections have regularly been reported worldwide during the SARS-CoV-2 pandemic [96,125,127,128,129]. Although the specific effect of *Candida auris* infection on mortality of critically ill COVID-19 patients has not been formally addressed, in one of the first CAC documentations coming from India, mortality was higher in candidemic COVID-19 patients with *Candida auris* compared with those with non-*auris* species [96].

Like other IFIs, mucormycosis is associated with considerable mortality in COVID-19 patients, which is heavily dependent on the site of infection. Additionally, major long-term sequelae are frequently reserved for survivors. In the largest study published so far, the investigators retrospectively analyzed the outcomes of 187 hospitalized patients with CAM in India [102]. A 3-month case fatality rate of 45.7% was estimated, which was not significantly different from that of another group of non-COVID patients diagnosed with mucormycosis during the same period. In a multivariate logistic regression analysis performed in the entire study population, older age, pulmonary mucormycosis, cerebral dissemination of rhino-orbital infection and ICU stay were all identified as risk factors for mortality. On the other hand, sequential antifungal combination treatment with amphotericin B plus a triazole (posaconazole or isavouconazole), although not recommended by current guidelines [130], was independently linked with a better prognosis. Recently, an international group of experts initiated by ECMM and ISHAM studied 80 published and unpublished cases of CAM from 18 countries [131]. A very similar all-cause mortality rate of 48.8% was found. Mortality was relentlessly high in those with pulmonary and disseminated mucormycosis (81% versus only 24.3% in isolated rhino-orbital disease). Again, central nervous system involvement in patients with rhino-orbital mucormycosis was shown to confer worse prognosis (59.1% mortality in this subgroup) and long-term disability in the form of vision loss in survivors (46.3% among survivors, all with cerebral involvement). Surgical therapy in combination with antifungals was demonstrated to improve survival in patients with isolated rhino-orbital infection. A high level of clinical suspicion allowing early identification of possible mucormycosis and timely initiation of amphotericin combined with aggressive surgical resection and debridement, when feasible, remain the mainstay of mucormycosis management, irrespective of COVID-19 status [130].

## 6. Conclusions

There is increasing evidence for the association between COVID-19 and IFIs. Emerging data demonstrate that the clinical course of COVID-19 can be complicated by a variety of fungal super-infections leading to unfavorable outcomes. It must be noted that critically ill COVID-19 patients have a number of risk factors predisposing them to fungal infections. The unique pathogenicity of SARS-CoV-2 with the deranged immune response it provokes, the well-known effects of immunosuppressant treatments and finally the causal relationships, synergistic effects or independent risk factors of critical illness compose the canvas of vulnerability for IFIs. The burden of fungal infections is largely not estimated. Studies based on histopathological confirmation are needed to improve our knowledge on the extent of the problem.

## Figures and Tables

**Figure 1 jcm-11-02017-f001:**
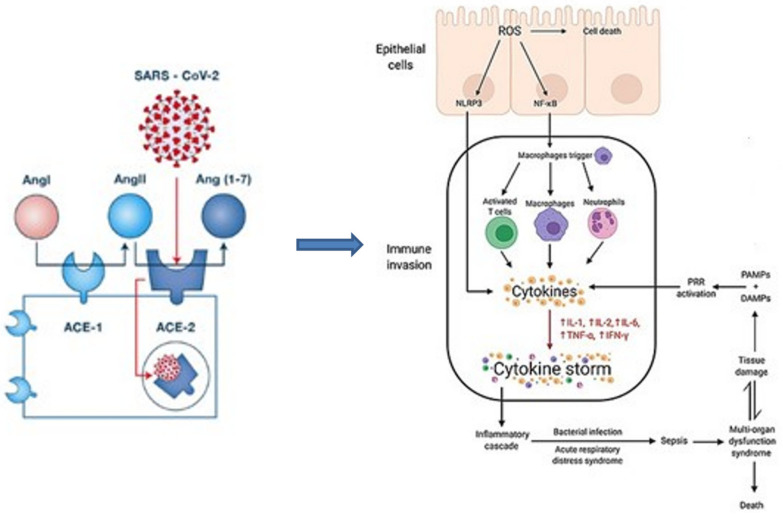
SARS-CoV2 spike protein binds to angiotensin converting enzyme 2 (ACE2) receptor of epithelial cells and type 2 pneumocytes, thus allowing viral entry. The release of danger-associated molecular patterns (DAMPs) by dying or damaged cells ignites an immune response and a cascade of inflammation, which in turn leads to tissue damage. This extensive lung damage may lead to higher vulnerability to invasive fungal infections.

**Table 1 jcm-11-02017-t001:** Risk factors for fungal infections.

CAPA/Invasive *Aspergillus* Tracheobronchitis
1. High/long dose of corticosteroids;
2. Underlying structural lung disease;
3. Host factors, such as neutropenia, allogeneic stem cell transplant, immunosuppression, inherited severe immunodeficiency;
4. Intubation and mechanical ventilation;
5. Cancer/chemotherapy;
6. Azithromycin (PMID: 33316401)/broad spectrum antibiotics;
7. Severe lung damage due to COVID-19.
CAC
1. Prolonged hospital stay;
2. Mechanical ventilation;
3. Central venous catheters;
4. Surgical procedures;
5. Broad-spectrum antibiotics.
MAC
1. Diabetes, diabetic ketoacidosis

CAPA: COVID-19-associated pulmonary aspergillosis; CAC: COVID-19-associated candidemia; MAC: Mycobacterium Avium Complex.

**Table 3 jcm-11-02017-t003:** Invasive Aspergillosis in COVID-19 patients (case reports and hematology patient case series excluded).

Literature	Trial Design and Population	Diagnostic Criteria Used	CS Used	CS Length	Other IST	Comorbidities	IA Incidence	Time to IA Dx	Mortality
Alanio A. et al. [87] France	P, O, *n* = 27 ICU pts, 9/27 CAPA, med age 63 [IQR 56–71]	EORTC-MSGERC or IAPA + ser β-D-glucan and qPCR (serum or pulm specimens)	6/9 pts: dexa IV 20 mg/d (D1–5) then 10 mg/d (D6–10). 2/9 pts on prev GCS	10 ds	NM	HPN more frequent in IPA (7/9 vs. 6/18, *p* = 0·046)	Probable IPAs: (4%) putative IPAs: 30%	NM	4/9
van Arkel A.E. et al. [88] Netherlands	O, *n* = 31 ICU pts on MV	*A. fumigatus* 5/6, A. Ag GM (+) BAL fluid: 3/6	3/6 pts: CS before IPA Dx, dose < 0.3 mg/kg/d	<3 wks	No	3/6 Pre-existing lung disease	6/31 (19.35%) presumed IPA	Sx onset—IPA: med 11.5 ds (8–42). ICU admis–IPA: med 5 ds (3–28)	66.7% died, med 12 ICU ds (11–20)
Bartoletti et al. [89] Italy	P, MC, *n* = 822	CAPA	MP 1 mg/kg	5–7 ds	TOCI		27.7%	Intub-CAPA: med 4ds (2–8). sx onset-CAPA: med 14ds (11–22)	↑↑ ICU mortality after adj for age, RRT, admis severity scores
Benedetti et al. [90] Argentina	*n* = 5 ICU pts	IAPA or EORTC-MSGERC serum markers, or AspICU	5/5 CS (<0.3 mg/kg)	<3wks	No	1/5 hematologic malignancy 2/5 diabetes		Sx onset–CAPA: 22 ds (13–52). ICU admis-CAPA: med 12 ds	1/5 died (rest still on MV)
Delliere et al. [92] France	R, O, MC, *n* = 360 ICU pts; 108 pts sampled on deterioration. 1 SOT. 1 myeloma	EORTC/MSGERC CAPA	NM	NM	Sarilumab 1 pt, eculizumab 6 pts, toci 4 pts	Azithromycin (>3 ds) and prob IPA (OR 3.1, 95% CI, 1.1–8.5, *p* = 0.02). HD dexa and IPA: 11.5% vs. 28.6%, (*p* = 0.08), cumul dose ≥100 mg and IPA (OR 3.7, 95% CI 1.0–9.7).	5.7% in ICU pts 8.5% in MV pts 19.4% in deteriorated pts	Sx onset- IPA: 16 ds (10–23) ICU admis—IPA: 6 ds (1–15)	IPA pts vs. non-IPA: 71.4% vs. 36.8%, *p* < 0.01).
Dupont et al. [91] France	R, 153 ICU pts screened for fungi; 106 PCR SARS-Co-V2 (+)	AspICU + serum/BAL GM	37% CS	short time	NM	HTN 36.8%, DM 36.8%, TB/COPD/asthma 36.8%	17.9% putative IPA	MV-CAPA: 6 ds	42%
Fekkar A. et al. [24] France	R, SC, *n* = 145 COVID-19 ICU MV pts screened for fungal superinfection; 54% on ECMO	EORTC/MSGERC, Mycology lab (microscopy, cultures, PCR respir samples and serum for *Aspergillus*, PJP, mucorales, GMI, β-D-glucan	Long-term (>3 wks) CS before COVID-19 and IFI (OR, 8.55; IQR, 6.8–10.3; *p* = 0.01), CS for COVID-19 (dexa 20 mg/d × 10 ds) no IFI	10 ds	6 Toci 3 saril 1 anti-IL1	100% MV, 68% ↑BW, 57% HTN, 32% DM, 14% preexisting immunosuppression	4.8% prob/ putat IFI (1 fusarium case), 17.2% colonization	ICU admis-IFI: med 7 ds (IQR, 2–56)	Survival 74.5%
Fortarezza et al. [94] Italy	*n* = 45 COVID-19 autopsies	Histology	CS: 88% of CAPA vs. 54% non-CAPA CS: 12/28 pts 1st wave vs. 16/17 pts 2nd wave	NM	No Toci No antiIL-1	7/9 ICU 7/9 HTN 3/9 COPD	20% proven CAPA, 1st wave 2/28 vs. 2nd wave 7/17	NM	NA
Janssen et al. [102] Belgium, Netherlands, France	O, MC, 2 ICU cohorts: N1 = 512 N2 = 304	ECCM/ISHAM	CS use not more prevalent in CAPA groups vs. non-CAPA	NM	Other IST < 90 ds before ICU admis	CAPA vs. nonCAPA: COPD 19% vs. 8% (*p* = 0.042). HIV (AIDS) 7% vs. 0.4% (*p* = 0.011)	10–15%	ICU admis to CAPA: 6 ds (IQR 3–9)	43–52%
Lamoth et al. [109] Switzerland	*n* = 80 ICU MV pts	IAPA	NM	NA	Toci—IPA Dx: 4 ds	No pt had any predisposing factors acc to EORTC/MSG	3.8% 1 probable 2 putative	COVID dx- IPA: med 9 ds, ICU admis-IPA: 6 ds, MV start-IPA: med 5 ds	1/3 died
Marr et al. [20] Spain, USA	R, MC *n* = 20 CAPA	*Aspergillus* recovery in BAS, sputum, BAL or GMI ≥ 1, imaging	NM Systemic and inh CS most common IST associated with CAPA	NM	NM	Age HTN Pulm dis underlying immunosuppressive disease/drugs	NA	Sx onset-CAPA: med 11 ds, ICU admis-CAPA: 9 ds	NM
Meijer et al. [93] Netherlands	SC, P, 1st wave: 33 MV ICU pts vs. 2nd wave: 33 MV ICU pts	2020 ECMM/ISHAM	All CAPA pts in 2nd wave on CS: Dexa 6 mg	10 ds	no	CVD 4/13 DM 3/13 HTN 2/13 COPD 1/13 ARF 1/13	1st vs. 2nd wave poss and prob CAPA: 15.2% vs. 25% (*p* = 0.36) In total: 19.7%	NM	40–50% mortality in both groups
Obata R. et al. [95] USA	R, 226 COVID-19 hosp pts, 57 on CS vs. 169 no-CS	NM	Dexa (48/57), P (6/57), MP (1/57), MP + P 1/57, HC 1/57	Max 10ds	20/57 Toci	NM	CAPA in CS vs. no-CS: 5.3% vs. 0.6% CAPA in toci vs. no-toci: 5% vs. 5.4%	NM	NM
Prattes et al. [111] Europe, USA	MC, P, MN 592 COVID-19 ICU pts	2020 ECMM/ISHAM	Majority on GCS	NM	Toci	Age MV Toci	Proven: 1.9%, Prob 13.5% poss: 3% No-CAPA: 81.6%	ICU admis-CAPA: 8 ds (0–31)	Survival in CAPA pts vs. non-CAPA: 29% vs. 57%
Rutsaert et al. [99] Belgium	*n* = 20 MV pts med 66 yo (56–77)	AspICU	1/7 CS (pemphigus)	NM	NM	4/7 DL 2/7 obesity 3/7 DM 3/7 HTN	7/20 (35%) proven IPA	Sx onset—IPA: 11–23 ds	4/7 died
Van Biesen et al. [100] Netherlands	42 MV ICU pts (9 IPA vs. 33 non-IPA)	AspICU + GMI ≥ 1	No CS	NA	NM	1/9 SOT COPD and asthma more common in IPA group	9/42	NM	22% IPA vs. 15.1% non-IPA (*p* = 0.6)
White et al. [30] UK	MC, P *n* = 137 ICU pts screened for IFI	AspICU, IAPA, CAPA	12/25 different CS	N/M	no	12/25 CRD 8/25 HTN 6/25 DM 6/25 obesity 5/25 CA	14.1% CAPA	ICU admiss- (+) Aspergillus tests: 8 ds (0–35)	CAPA mortality 57.9% depending on appropriate Tx

admis = admission, ARF = Acute Renal Failure, BAS = Bronchial aspirate, BSI = Blood-stream infections, BW = body weight, CFR = Case Fatality Rate, CS = corticosteroids, CRD = Chronic Respiratory Disease, cumul = cumulative, CVD = cardiovascular disease, DL = dyslipidemia, Ds = days, dexa = dexamethasone, DM = diabetes mellitus, Dx = diagnosis, ECMO = Extra-corporeal membrane oxygenation, GM = galactomannan, GM = Galactomannan, GMI = galactomannan index, HD = high dose, HTN = Hypertension, IA = Invasive Aspergillosis, IFI = invasive fungal infection, intub = intubation, IST = immunosuppressive therapy, MC = Multicenter, med = median, MN = multi-national, MP = methylprednisolone, MV = mechanically ventilated, m = median, *n* = number of patients, NA = not applicable, NM = not mentioned, O = observational, OR = odds Ratio, P = Prospective, prev = previous, prob = probable, put = putative, R = Retrospective, resp sampl on deterior = respiratory sampling on deterioration, ROM = Rhino-orbital mucormycosis, saril = sarilumab, SC = single center, Sx = symptom, Toci = tocilizumab, unclass = unclassified.

**Table 4 jcm-11-02017-t004:** *Candida* infections in COVID-19 patients treated with steroids.

Literature	Trial Design	CS Used	CS Length	Other IST	Comorbidities/ Risk Factors	Candida Infection Incidence	Time to CAC Dx	Mortality
Antinori et al. [112] Italy	*n* = 43 severe COVID-19 pts; 3/43 candidemia	NM	NM	Toci	TPN (3/3), antibiotics (2/3), toci (3/3)	6.9% BSI	Toci last dose—CAC: med 13 ds	Still hospitalized on publication
Chowdhary et al. [97] India	*n* = 596 COVID-19 ICU pts, 420 MV, 15 *Candida* BSI	NM	NM	NM	↑ ICU LOS, HTN, DM, CKD, CS (10/15)	2.5% BSI	Admis-CAC: 10–12 ds	53% (60% for *C. auris*)
Ho et al. [98] USA	R, O, *n* = 4313 hospitalised, 574 (13.3%) received CS	MP > P > dexa	6.34–9.53 ds	Toci	HTN 35.4% DM23.4% CKD 13%	BSI	NM	
Obata et al. [95] USA	R, 226 COVID-19 hosp pts, 57 on CS vs. 169 no-CS	See Table 1	Max 10ds	20/57 Toci	NM	CAC in CS vs. no-CS: 7% vs. 0% CAC in toci vs. no-toci: 15% vs. 2.7%	NM	NM
Riche et al. [96] Brazil	R, candidemia incidence between COVID and non-COVID inpatients	MP > dexa > P	2–13 ds	No	HD CS CVC 90.9% ICU pts (72.7%)	×10 increase in candidemia	ICU admis-CAC: 0–22 ds	72.7% following CS use
Seagle et al. [36] USA	Candidemia in COVID-19 and non-COVID-19 pts, surveillance data	NM	NM	Toci more likely among pts with COVID-19 compared to non-COVID-19 pts	Candidemia RF in non-COVID pts: LD, malignancy, prior surgeries CAC each >1.3 times more common: ICU, MV, CVC, CS, IST. Common RF in COVID-19 pts: DM, obesity.	CS within 30 ds of CAC: ×2 vs. non-COVID-19 pts	SARS-CoV-2 (+) test-*Candida* culture: med 15 ds ([IQR]: 8–21 days)	CAC: ×2 mortality (62.5%) vs. candidemia in non-COVID-19 pts (32.1%)
Segrelles-Calvo [114] et al. Spain	O, P, *n* = 218 ICU pts	MP	1–10 ds	Toci-CAC: RR 1.378, *p* = 0.05. Toci + MP/dexa (*p* = 0.01)	Malignancies more common in COVID-19 with candida co-infection. ICU, TPN, CVC, ↑LO ICU stay	14.4% (+) Candida tests		

N = number, admis = admission, ARF = Acute Renal Failure, BAS = Bronchial aspirate, BSI = Blood-stream infections, BW = body weight, CFR = Case Fatality Rate, CS = corticosteroids, CRD = Chronic Respiratory Disease, cumul = cumulative, CVD = cardiovascular disease, DL = dyslipidemia, Ds = days, dexa = dexamethasone, DM = diabetes mellitus, Dx = diagnosis, ECMO = Extra-corporeal membrane oxygenation, GM = galactomannan, GM = Galactomannan, GMI = galactomannan index, HD = high dose, HTN = Hypertension, IA = Invasive Aspergillosis, IFI = invasive fungal infection, intub = intubation, IST = immunosuppressive therapy, MC = Multicenter, med = median, MN = multi-national, MP = methylprednisolone, MV = mechanically ventilated, m = median, *n* = number of patients, NA = not applicable, NM = not mentioned, O = observational, OR = odds Ratio, P = Prospective, prev = previous, prob = probable, put = putative, R = Retrospective, resp sampl on deterior = respiratory sampling on deterioration, ROM = Rhino-orbital mucormycosis, saril = sarilumab, SC = single center, Sx = symptom, Toci = tocilizumab, unclass = unclassified.

## Data Availability

Not applicable.

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
