# Peer review of "Fungal Infections in Critically Ill COVID-19 Patients: Inevitabile Malum"

_jcm, 2022, doi:10.3390/jcm11072017_

Round 1

Reviewer 1 Report

Dear Editor and authors,

The review article entitled “Fungal infections in critically ill COVID-19 patients: inevitabile malum” brings relevant information regarding the incidence of fungal infections and their relationship with COVID-19. The manuscript brings a robust search of bibliography and discuss the results found of a variety of studies. It will certainly contribute to clarify the importance of fungal diseases as a secondary infection related to COVID-19.

Some suggestions and alterations are listed below.

Overall comments:

1) A careful grammar and English revision should be performed in the text. Please see below some mistakes I identified during the reading:

- Please italicize all microorganism names, all of them appeared to me as non-italicized words. In addition, please put the letter C of Candida in capital letters, such as in “candida albicans”, “candida glabrata” and “candida auris”.

- COVID-19 sometimes were written with no capital letters, sometimes I found “COVID-29”. Please check it along the text.

- Please check the use of comma along the text, sometimes it seems to be in a wrong place, which make it harder to read and understand the information.

Abstract:

1) the first word “patients” is in bold format. Please correct it.

2) Please change “DC8+” to “CD8+”

Introduction:

1) Some abbreviations are not defined at the very beginning of the introduction, such as SARS, MERS and ARDS. Their definition would help a better comprehension of the text.

2) Second paragraph: what do you mean by superinfections? I suggest to briefly enumerate the reasons why these secondary infections are considered superinfections.

3) Second paragraph: Please change “DC8+” to “CD8+”

4) I miss an explanation of why Italian ICU presented 20 times higher incidence of fungal infections than European ICUs. The authors should briefly explore this information to improve it in the text.

Topic 2. Prevalence of invasive fungal infections in critically ill COVID-19 patients:

1) I suggest the authors to elaborate a table summarizing the different incidence rates found on each study. Also, the incidence of each fungal species could be added to this table. These data shown in a table would make it easier for the readers to visualize the information.

Topic 3. Pathophysiology and risk factors:

1) I did not receive Figure 1 to appreciate. I do not know if it was not submitted by the authors or if it did not reach me in the review process. Please make sure that the Figure is available or remove the citation in the text.

2) What is the meaning of EORTC/MSGERC? Please define the abbreviations in the text.

3) Since topic 4 specifically discuss the role of immunomodulation, I would suggest the authors to consider the removal of the topic 3.3 and mention before the topic 3.1 that this information will be discussed separately in the topic 4.

Author Response

                                                                                                                           Athens, March 28, 2022

Dear editor,

We would like to thank the reviewers for their careful reading and constructive criticism, which helped towards a substantial improvement of the manuscript.

We address the reviewers’ comments and issues raised point by point below.

Reviewer 1

Overall comments:

1) A careful grammar and English revision should be performed in the text. Please see below some mistakes I identified during the reading:

- Please italicize all microorganism names, all of them appeared to me as non-italicized words. In addition, please put the letter C of Candida in capital letters, such as in “candida albicans”, “candida glabrata” and “candida auris”.

- COVID-19 sometimes were written with no capital letters, sometimes I found “COVID-29”. Please check it along the text.

- Please check the use of comma along the text, sometimes it seems to be in a wrong place, which make it harder to read and understand the information.

We carefully read the manuscript and corrected grammar and language issues. We also italicized all microorganism names and corrected commas and the reference of COVID-19 to be uniform along the manuscript.

Abstract:

1) the first word “patients” is in bold format. Please correct it.

2) Please change “DC8+” to “CD8+”

 We corrected the points indicated by the reviewer.

Introduction:

  • Some abbreviations are not defined at the very beginning of the introduction, such as SARS, MERS and ARDS. Their definition would help a better comprehension of the text.

We added the full description of the abbreviations  

  • Second paragraph: what do you mean by superinfections? I suggest to briefly enumerate the reasons why these secondary infections are considered superinfections.

The reviewer’s comment is accurate. We erased the word “superinfections” which is misleading and added a list of reasons predisposing to secondary infections.

  • Second paragraph: Please change “DC8+” to “CD8+”

We corrected it.

4) I miss an explanation of why Italian ICU presented 20 times higher incidence of fungal infections than European ICUs. The authors should briefly explore this information to improve it in the text.

 We added an explanation in the text based on the original justification by the authors in the referenced article.

Topic 2. Prevalence of invasive fungal infections in critically ill COVID-19 patients:

1) I suggest the authors to elaborate a table summarizing the different incidence rates found on each study. Also, the incidence of each fungal species could be added to this table. These data shown in a table would make it easier for the readers to visualize the information.

 A Table (Table 2) summarizing the different incidence rates has been adding according to reviewer’s suggestion.

Topic 3. Pathophysiology and risk factors:

  • I did not receive Figure 1 to appreciate. I do not know if it was not submitted by the authors or if it did not reach me in the review process. Please make sure that the Figure is available or remove the citation in the text.

Figure 1 is embedded in the text. It was mistakenly not uploaded in the original submission. We thank the reviewer for his/her comment.

  • What is the meaning of EORTC/MSGERC? Please define the abbreviations in the text.

The definition of the abbreviation is added in the test.

3) Since topic 4 specifically discuss the role of immunomodulation, I would suggest the authors to consider the removal of the topic 3.3 and mention before the topic 3.1 that this information will be discussed separately in the topic 4.

The modification suggested by the reviewer has been made.

Reviewer 2 Report

This is a well-documented review article that sought to shed some light on whether SARS-CoV-2 inherently predisposes to fungal infections in the critically ill patients or the immunosuppressive therapy constitutes the igniting factor for invasive mycoses. This review aimed to identify the prevalence, risk factors, and mortality associated with IFIs in mechanically ventilated patients with COVID-19.

Although well documented and interesting a few issues should be addressed.

Please see the enclosed PDF for details

Author Response

                                                                                                                           Athens, March 28, 2022

Dear editor,

We would like to thank the reviewers for their careful reading and constructive criticism, which helped towards a substantial improvement of the manuscript.

We address the reviewers’ comments and issues raised point by point below.

Reviewer 2

  1. We split the section to subdivisions as suggested by the reviewer.
  2. We placed the section “Pathophysiology and risk factors” after Introduction, according to reviewer’s suggestion.
  3. Furthermore, in the second paragraph of this section we added the suggested refence.
  4. We turned the text into a Table (Table 1) to be better understandable.
  5. We also split section 4 into subdivisions as suggested for the better understanding of the content.

Round 2

Reviewer 2 Report

The manuscript has been improved